# Use of Whole-Genome Sequencing to Predict *Mycobacterium tuberculosis* Complex Drug Resistance from Early Positive Liquid Cultures

Xiaocui Wu,[a] Guangkun Tan,[b] Wei Sha,[c] Haican Liu,[d] Jinghui Yang,[a] Yinjuan Guo,[a] Xin Shen,[e] Zheyuan Wu,[e] Hongbo Shen,[c] Fangyou Yu[a]

[a]Department of Clinical Laboratory, Shanghai Pulmonary Hospital, School of Medicine, Tongji University, Shanghai, People's Republic of China
[b]Department of Clinical Laboratory, Shanghai University of Traditional Chinese Medical Attached Shuguang Hospital, Shanghai, People's Republic of China
[c]Department of Tuberculosis, Shanghai Pulmonary Hospital, School of Medicine, Tongji University, Shanghai, People's Republic of China
[d]State Key Laboratory for Infectious Diseases Prevention and Control, Collaborative Innovation Center for Diagnosis and Treatment of Infectious Diseases, National Institute for Communicable Disease Control and Prevention, Chinese Center for Disease Control and Prevention, Beijing, People's Republic of China
[e]Shanghai Municipal Center for Disease Control and Prevention, Shanghai, People's Republic of China

Xiaocui Wu and Guangkun Tan contributed equally to this article. Author order was determined in order of decreasing seniority.

**ABSTRACT** Our objective was to evaluate the performance of whole-genome sequencing (WGS) from early positive liquid cultures for predicting *Mycobacterium tuberculosis* complex (MTBC) drug resistance. Clinical isolates were obtained from tuberculosis patients at Shanghai Pulmonary Hospital (SPH). Antimicrobial susceptibility testing (AST) was performed, and WGS from early Bactec mycobacterial growth indicator tube (MGIT) 960-positive liquid cultures was performed to predict the drug resistance using the TB-Profiler informatics platform. A total of 182 clinical isolates were enrolled in this study. Using phenotypic AST as the gold standard, the overall sensitivity and specificity for WGS were, respectively, 97.1% (89.8 to 99.6%) and 90.4% (83.4 to 95.1%) for rifampin, 91.0% (82.4 to 96.3%) and 95.2% (89.1 to 98.4%) for isoniazid, 100.0% (89.4 to 100.0%) and 87.3% (80.8 to 92.1%) for ethambutol, 96.6% (88.3 to 99.6%) and 61.8% (52.6 to 70.4%) for streptomycin, 86.8% (71.9 to 95.6%) and 95.8% (91.2 to 98.5%) for moxifloxacin, 86.5% (71.2 to 91.5%) and 95.2% (90.3 to 98.0%) for ofloxacin, 100.0% (54.1 to 100.0%) and 67.6% (60.2 to 74.5%) for amikacin, 100.0% (63.1 to 100.0%) and 67.2% (59.7 to 74.2%) for kanamycin, 62.5% (24.5 to 91.5%) and 88.5% (82.8 to 92.8%) for ethionamide, 33.3% (4.3 to 77.7%) and 98.3% (95.1 to 99.7%) for para-aminosalicylic acid, and 0.0% (0.0 to 12.3%) and 100.0% (97.6 to 100.0%) for cycloserine. The concordances of WGS-based AST and phenotypic AST were as follows: rifampin (92.9%), isoniazid (93.4%), ethambutol (89.6%), streptomycin (73.1%), moxifloxacin (94.0%), ofloxacin (93.4%), amikacin (68.7%), kanamycin (68.7%), ethionamide (87.4%), para-aminosalicylic acid (96.2%) and cycloserine (84.6%). We conclude that WGS could be a promising approach to predict MTBC resistance from early positive liquid cultures.

**IMPORTANCE** In this study, we used whole-genome sequencing (WGS) from early positive liquid (MGIT) cultures instead of solid cultures to predict drug resistance of 182 *Mycobacterium tuberculosis* complex (MTBC) clinical isolates to predict drug resistance using the TB-Profiler informatics platform. Our study indicates that WGS may be a promising method for predicting MTBC resistance using early positive liquid cultures.

**KEYWORDS** drug-resistant tuberculosis, whole-genome sequencing, antimicrobial susceptibility testing, early positive liquid cultures

Address correspondence to Hongbo Shen, hbshen@tongji.edu.cn, or Fangyou Yu, wzjxyfy@163.com.

The authors declare no conflict of interest.

Tuberculosis (TB) caused by *Mycobacterium tuberculosis* complex (MTBC) infection is still an important infectious disease and a serious public health and social problem. According to a World Health Organization (WHO) report on the year 2021, there were an estimated 9.9 million new cases of tuberculosis worldwide and 1.3 million deaths. China is a high-burden country for TB, second only to India (1). There were 842,000 new TB cases, an increase of 9,000 compared to the number of cases in 2019 in China (1). Inappropriate treatment can delay treatment and lead to the development of acquired drug resistance. Therefore, timely and accurate detection of MTBC drug susceptibility is essential for the treatment and control of TB.

The diagnosis, treatment, monitoring, and control of MTBC depend on rapid and accurate antimicrobial susceptibility testing (AST). Phenotypic AST, the gold standard, is a time-consuming, complex operation with high biosafety requirements. With in-depth research on the molecular mechanisms of drug resistance in MTBC, molecular diagnostic techniques for rapid diagnosis of drug-resistant tuberculosis by detecting drug-resistant gene mutations have received increasing attention. Molecular diagnosis can greatly improve detection efficiency, but there are differences in the diagnostic efficiencies of different kits, the coverage of antituberculosis drugs, and the corresponding drug-resistance gene loci (2). Current commercial molecular diagnostic methods can only a detect limited number of mutation sites. For example, GeneXpert MTB/RIF (Cepheid) can only detect mutations in an 81-bp rifampin-resistance determining region (RRDR) of *rpoB*, and GenoType MTBDRplus (Hain Lifescience) can only detect mutations in the RRDR of *rpoB* and the isoniazid-resistance genes *katG* and *inhA*. The Xpert MTB/XDR (Cepheid) can be used to simultaneously amplify eight genes and promoter regions in MTBC, and it analyzes melting temperatures ($T_m$) to identify mutations associated with isoniazids, fluoroquinolones, ethionamide, amikacin, kanamycin, and capreomycin resistance (3, 4). Heterogeneous resistance cannot be detected, and synonymous mutations or silent mutations may also be incorrectly reported as resistance (5).

Compared with these other molecular biologic approaches for testing TB drug susceptibility, whole-genome sequencing (WGS) has the advantages of identifying all existing mutations and effectively identifying both synonymous and silent mutations (6, 7). A number of studies have demonstrated that culture-based WGS can be used to detect drug resistance in MTBC, especially against rifampin and isoniazid (8–14). The application of WGS in predicting MTBC drug resistance directly from clinical isolates is limited, due to the small amount of MTBC and the high technical requirements for sequencing (15–17). Previous studies using WGS to predict drug resistance of MTBC have mostly used pure cultures after solid cultures. Bactec MGIT 960 (Becton, Dickinson, Cockeysville, MD, USA) has obvious advantages over smear and Löwenstein-Jensen (LJ) cultures in terms of positive detection rate and culture time, shortening sample turnaround time (TAT) and increasing the positive rate (18, 19). In this study, the early Bactec MGIT 960-positive liquid cultures were directly used for WGS to predict the drug resistance of MTBC.

## RESULTS

**Drug-resistance patterns of MTBC isolates.** A total of 182 clinical isolates were enrolled in this study. Among the 29 isolates excluded, 21 were sequenced at a depth of less than 20×, 2 were phenotypic AST-contaminated, and 6 had poor growth for phenotypic AST. Table 1 shows the drug resistance profiles of all 182 clinical MTB isolates according to phenotypic AST results. Of these, 57.1% (104/182) were resistant to at least one drug, 47.3% (86/182) were resistant to any first-line drug, and 46.2% (84/182) were resistant to any second-line drug. The total resistance rates were as follows: rifampin (68 isolates), 37.4%; isoniazid (78), 42.9%; ethambutol (33), 18.1%; streptomycin (59), 32.4%; moxifloxacin (38), 20.9%; ofloxacin (37), 20.3%; amikacin (6), 3.3%; kanamycin (8), 4.4%; ethionamide (8), 4.4%; para-aminosalicylic acid (6), 3.3%; and cycloserine (28), 15.4%. Of all TB isolates, 32.4% (59/182) were diagnosed as multidrug-resistant TB (MDR-TB, resistant at least to isoniazid and rifampin).

**Mutations associated with drug resistance.** Mutations associated with resistance to the antituberculosis drugs in MTBC were identified by WGS (Table 2). The amino

**TABLE 1** Drug resistance profile of all 182 clinical MTBC isolates according to phenotypic AST results

| Resistance pattern | No. (%) of strains |
|---|---|
| Susceptible to all drugs | 78 (42.9%) |
| Resistant to any drug | 104 (57.1%) |
| | |
| Resistant to any first-line drug | 86 (47.3%) |
| Rifampin | 68 (37.4%) |
| Isoniazid | 78 (42.9%) |
| Ethambutol | 33 (18.1%) |
| | |
| Resistant to second-line drugs | 84 (46.2%) |
| Streptomycin | 59 (32.4%) |
| Moxifloxacin | 38 (20.9%) |
| Ofloxacin | 37 (20.3%) |
| Amikacin | 6 (3.3%) |
| Kanamycin | 8 (4.4%) |
| Ethionamide | 8 (4.4%) |
| Para-aminosalicylic_acid | 6 (3.3%) |
| Cycloserine | 28 (15.4%) |
| | |
| Multidrug-resistant (MDR) | 59 (32.4%) |

acid mutations and frequency associated with drug resistance in this study are provided in Table S1 in the supplemental material. Of the 77 isolates with rifampin resistance-associated mutations, 13 had more than one mutation. Eleven isolates contained mutations outside the RRDR. Among them, 2 isolates had only the *rpoB* Leu464Met mutation, 2 had mutations in both the *rpoC* gene and the RRDR, and the rest had more than one mutation in and outside the RRDR. A total of 76 (41.8%) isolates had mutations in genes associated with resistance to isoniazid, including *katG* (*n* = 66), the promoter of *fabG1* (*n* = 11), *ahpC* (*n* = 5), and *inhA* (*n* = 1). Eight isolates had two isoniazid resistance-associated mutations. One of these had mutations at two different *katG* sites, and the remaining seven had mutations in different genes. The most common mutation was *katG* Ser315Thr (*n* = 61). There were 52 strains with genetic ethambutol

**TABLE 2** Mutations associated with resistance to antituberculosis drugs in MTBC, as identified by WGS

| Drug | Gene | No. of isolates |
|---|---|---|
| Rifampin | *rpoB* | 77 |
| | *rpoC* | 2 |
| Isoniazid | *katG* | 66 |
| | *fabG1* | 11 |
| | *ahpC* | 5 |
| | *inhA* | 1 |
| Ethambutol | *embA* | 9 |
| | *embB* | 47 |
| Streptomycin | *rrs* | 73 |
| | *rspl* | 55 |
| | *gid* | 4 |
| Fluoroquinolones | *gyrA* | 36 |
| | *gyrB* | 5 |
| Amikacin | *rrs* | 63 |
| Kanamycin | *rrs* | 63 |
| | *eis* | 2 |
| Ethionamide | *ethA* | 17 |
| | *fabG1* | 9 |
| | *inhA* | 1 |
| Para-aminosalicylic acid | *thyX* | 3 |
| | *thyA* | 2 |

resistance. Nine isolates had *embA* mutations and 47 isolates had *embB* mutations. Among these, 4 isolates had both *embA* and *embB* mutations, and 2 isolates had different mutations at different *embB* sites. The most common mutations were *embB* Met306Val and Met306Ile, accounting for 25.0% (13/52) and 21.2% (11/52) of the ethambutol-resistant strains, respectively. Streptomycin resistance-associated genes were *rrs*, *rspL*, and *gid* in 73, 55, and 4 isolates, respectively. Among these, 26 isolates had both *rspL* and *rrs* mutations, and 1 isolate had mutations in all three streptomycin resistance-associated genes. For fluoroquinolones, 36 isolates (92.3%) showed mutations in the *gyrA* gene, with the mutation Asp94Gly being most common ($n = 19$), and rare mutations at codon 89 in 1 isolate. Five isolates had mutations in *gyrB*, namely, Asp461Asn, Asp461Ala, Ala504Thr, Glu501Asp, and Ser447Phe. All 63 amikacin and kanamycin genotype-resistant isolates had mutations in the *rrs* gene, the most common mutations were 1,402 C→A ($n = 59$) and 1,484 G→T ($n = 49$). In addition, two kanamycin genotype-resistant isolates had mutations in the *eis* gene. Ethionamide resistance-associated mutations were identified in 25 isolates, including *ethA* ($n = 17$), the promoter of *fabG1* ($n = 9$) and *inhA* ($n = 1$). Among them, one isolate had both *ethA* and the promoter of *fabG1* mutations, and another had the promoter of both *fabG1* and *inhA* mutations. Mutations of the para-aminosalicylic acid-resistance gene were only found in 5 isolates, while mutations of the cycloserine resistance gene were not found in this study.

**Agreement of phenotypic and genotypic AST.** Using phenotypic AST as the gold standard, we evaluated the ability of WGS, using the early Bactec MGIT 960-positive liquid cultures, to predict MTBC drug susceptibility for 11 drugs. Concordance, sensitivity, specificity, positive predictive value (PPV), and negative predictive value (NPV) for each drug are shown in Table 3. For 182 isolates tested, we found an average concordance of 85.6% across all 11 drugs, ranging from 68.7% (amikacin and kanamycin) to 96.2% (para-aminosalicylic acid). The sensitivity and specificity for WGS to predict MTBC drug susceptibility were, respectively, 97.1% (89.8 to 99.6%) and 90.4% (83.4 to 95.1%) for rifampin, 91.0% (82.4 to 96.3%) and 95.2% (89.1 to 98.4%) for isoniazid, 100.0% (89.4 to 100.0%) and 87.3% (80.8 to 92.1%) for ethambutol, 96.6% (88.3 to 99.6%) and 61.8% (52.6 to 70.4%) for streptomycin, 86.8% (71.9 to 95.6%) and 95.8% (91.2 to 98.5%) for moxifloxacin, 86.5% (71.2 to 91.5%) and 95.2% (90.3 to 98.0%) for ofloxacin, 100.0% (54.1 to 100.0%) and 67.6% (60.2 to 74.5%) for amikacin, 100.0% (63.1 to 100.0%) and 67.2% (59.7 to 74.2%) for kanamycin, 62.5% (24.5 to 91.5%) and 88.5% (82.8 to 92.8%) for ethionamide, 33.3% (4.3 to 77.7%) and 98.3% (95.1 to 99.7%) for para-aminosalicylic acid, and 0.0% (0.0 to 12.3%) and 100.0% (97.6 to 100.0%) for cycloserine.

There were 121 isolates with inconsistent phenotypic and genotypic AST results for one or more drugs, of which only 39 cases had inconsistent phenotypic and genotypic resistance results for any first-line drugs (rifampin, isoniazid, and ethambutol). Of these, 103 isolates showed drug-resistance associated mutations but were susceptible by phenotypic AST. For these isolates, mutations are shown in Table 4. Conversely, 43 isolates showed no drug resistance-associated mutations, but phenotypic AST showed resistance to one or more drugs. For these isolates, the minimum inhibitory concentration (MIC) values are 4 $\mu$g/mL ($n = 1$) and >16 $\mu$g/mL ($n = 1$) for rifampin; 1 $\mu$g/mL ($n = 3$), 0.25 $\mu$g/mL ($n = 2$), 0.5 $\mu$g/mL ($n = 1$), and 4 $\mu$g/mL ($n = 1$) for isoniazid; 32 $\mu$g/mL ($n = 1$) and >32 $\mu$g/mL ($n = 1$) for streptomycin; 2 $\mu$g/mL ($n = 3$) and 1 $\mu$g/mL ($n = 2$) for moxifloxacin; 8 $\mu$g/mL ($n = 3$) and 2 $\mu$g/mL ($n = 2$) for ofloxacin; 10 $\mu$g/mL ($n = 1$) and >40 $\mu$g/mL ($n = 2$) for ethionamide; 4 $\mu$g/mL ($n = 3$) and 32 $\mu$g/mL ($n = 1$) for para-aminosalicylic acid; and 32 $\mu$g/mL ($n = 25$) and 64 $\mu$g/mL ($n = 3$) for cycloserine.

**Phylogenetic distribution of drug resistance.** A phylogenetic tree of 182 MTBC isolates indicating drug-resistance profiles and lineages with 81,622 high-confidence single-nucleotide polymorphisms (SNPs) is shown in Fig. 1. Of these, 164 isolates belong to lineage 2 (East Asian), 13 isolates belong to lineage 4 (Europe and America), 4 isolates belong to lineages 2 (East Asian) and 4 (Euro-American), and the remaining 1 is *Mycobacterium bovis*.

**TABLE 3** Whole-genome sequencing compared with phenotypic antimicrobial susceptibility testing in drug-resistance diagnosis of *Mycobacterium tuberculosis* complex[a]

| | No. of isolates | | | | | | | | |
| | Phenotypically resistant | | Phenotypically susceptible | | | | | | |
| | Genetically | | Genetically | | | | | | |
| Drugs | Resistant | Susceptible | Resistant | Susceptible | Concordance (%) | Sensitivity (%) | Specificity (%) | PPV | NPV |
|---|---|---|---|---|---|---|---|---|---|
| Rifampin | 66 | 2 | 11 | 103 | 92.9 | 97.1 (89.8–99.6) | 90.4 (83.4–95.1) | 85.7 (75.9–92.7) | 98.1 (93.3–99.8) |
| Isoniazid | 71 | 7 | 5 | 99 | 93.4 | 91.0 (82.4–96.3) | 95.2 (89.1–98.4) | 93.4 (85.3–97.8) | 93.4 (86.9–97.3) |
| Ethambutol | 33 | 0 | 19 | 130 | 89.6 | 100.0 (89.4–100) | 87.3 (80.8–92.1) | 63.5 (49.0–76.4) | 100.0 (97.2–100.0) |
| Streptomycin | 57 | 2 | 47 | 76 | 73.1 | 96.6 (88.3–99.6) | 61.8 (52.6–70.4) | 54.8 (44.7–64.6) | 97.4 (91.0–99.7) |
| Moxifloxacin | 33 | 5 | 6 | 138 | 94.0 | 86.8 (71.9–95.6) | 95.8 (91.2–98.5) | 84.6 (69.5–94.1) | 96.5 (92.0–98.9) |
| Ofloxacin | 32 | 5 | 7 | 138 | 93.4 | 86.5 (71.2–91.5) | 95.2 (90.3–98.0) | 82.1 (66.5–92.5) | 96.5 (92.0–98.9) |
| Amikacin | 6 | 0 | 57 | 119 | 68.7 | 100.0 (54.1–100.0) | 67.6 (60.2–74.5) | 9.5 (3.6–19.6) | 100.0 (97.0–100.0) |
| Kanamycin | 8 | 0 | 57 | 117 | 68.7 | 100.0 (63.1–100.0) | 67.2 (59.7–74.2) | 12.3 (5.5–22.8) | 100.0 (96.9–100.0) |
| Ethionamide | 5 | 3 | 20 | 154 | 87.4 | 62.5 (24.5–91.5) | 88.5 (82.8–92.8) | 20.0 (6.8–40.7) | 98.1 (94.5–99.6) |
| Para-aminosalicylic acid | 2 | 4 | 3 | 174 | 96.2 | 33.3 (4.3–77.7) | 98.3 (95.1–99.7) | 40.0 (5.27–85.3) | 97.8 (94.4–99.4) |
| Cycloserine | 0 | 28 | 0 | 154 | 84.6 | 0 (0–12.3) | 100.0 (97.6–100.0) | NA | 84.6 (78.5–89.5) |

[a]PPV, positive predictive value; NPV, negative predictive value.

**TABLE 4** Mutations and their frequencies in phenotypic drug-susceptible isolates with drug-resistance mutations

| Drug | Mutations | No. of isolates |
|---|---|---|
| Rifampicin | *rpoB* Asp435Tyr | 2 |
| | *rpoB* Leu464Met | 2 |
| | *rpoB* Leu430 | 1 |
| | *rpoB* His445Asn | 1 |
| | *rpoB* His445Cys | 1 |
| | *rpoB* Asp435Gly | 1 |
| | *rpoB* Leu452Pro | 1 |
| | *rpoB* Ser450Leu | 1 |
| | *rpoB* Leu464Met, Ser450Tyr | 1 |
| Isoniazid | *fabG1* −15 C→T | 2 |
| | *katG* Ser315Thr | 1 |
| | *ahpC* −48 G→A | 1 |
| | *ahpC* −54 C→T, *katG* 2023 2024del | 1 |
| Ethambutol | *embB* Met306Ile | 6 |
| | *embB* Gly406Asp | 4 |
| | *embA* 12 C→T | 3 |
| | *embB* His1002Arg | 2 |
| | *embB* Gln497Arg | 1 |
| | *embB* Asp354Ala | 1 |
| | *embB* Gly406Cys | 1 |
| | *embB* Ala388Gly, *embB* His312Arg, *embB* Leu359Ile | 1 |
| Streptomycin | *rrs* 799 C→T, *rrs* 888 G→A | 23 |
| | *rrs* 799 C→T | 8 |
| | *rrs* 888 G→A | 7 |
| | *rpsL* Lys88Arg, *rrs* 799 C→T, *rrs* 888 G→A | 2 |
| | *gid* 115del | 1 |
| | *gid* 326del | 1 |
| | *gid* 351del | 1 |
| | *rpsL* Lys43Arg, *rrs* 799 C→T | 1 |
| | *rrs* 462 C→T, *rrs* 888 G→A | 1 |
| | *rrs* 514 A→C | 1 |
| | *rrs* 517 C→T, *rrs* 799 C→T, *rrs* 888 G→A | 1 |
| Moxifloxacin | *gyrA* Asp94Gly | 2 |
| | *gyrA* Asp94Ala | 1 |
| | *gyrB* Ser447Phe | 1 |
| | *gyrA* Ala90Val | 1 |
| | *gyrB* Ala504Thr | 1 |
| Ofloxacin | *gyrA* Asp94Gly | 2 |
| | *gyrA* Asp94Ala | 1 |
| | *gyrB* Ser447Phe | 1 |
| | *gyrA* Ala90Val | 1 |
| | *gyrB* Ala504Thr | 1 |
| | *gyrB* Glu501Asp | 1 |
| Amikacin | *rrs* 1,402 C→A, *rrs* 1,484 G→T | 47 |
| | *rrs* 1,402 C→A | 9 |
| | *rrs* 1,401 A→G, *rrs* 1,402 C→A | 1 |
| Kanamycin | *rrs* 1,402 C→A, *rrs* 1,484 G→T | 46 |
| | *rrs* 1,402 C→A | 9 |
| | *rrs* 1,401 A→G, *rrs* 1,402 C→A | 1 |
| | *eis* -10G→A | 1 |
| Ethionamide | *fabG1* -15C→T | 5 |
| | *ethA* 504del | 1 |
| | *ethA* 752_753insG | 1 |
| | *ethA* 679del | 1 |

**TABLE 4** (Continued)

| Drug | Mutations | No. of isolates |
|---|---|---|
| | *ethA* Chromosome:g.4326679_4327538del | 1 |
| | *ethA* 1290del | 1 |
| | *ethA* 552_553insC | 1 |
| | *ethA* 1341del | 1 |
| | *ethA* 223del | 1 |
| | *ethA* 240del | 1 |
| | *ethA* 284_285insATCGA | 1 |
| | *ethA* 140del | 1 |
| | *ethA* 1299_1300insG | 1 |
| | *ethA* Chromosome:g.4327134_4327434del | 1 |
| | *ethA* 620_621insG | 1 |
| | *ethA* 491del | 1 |
| Para-aminosalicylic acid | *thyA* His75Asn | 2 |
| | *thyX* -16C→T | 1 |

## DISCUSSION

Since the complete genome sequence of MTBC was described in 1998, WGS has been widely used in the diagnosis, epidemic investigation, and drug and vaccine development for TB (20). Currently, WGS is not routinely used clinically to predict TB resistance in China due to its difficulties in standardization, interpretation of results, etc. With the sharing of genome-wide data, the in-depth research of drug resistance mechanisms, and the development of sequencing technology, WGS may completely change AST in conventional microbiology laboratories. It is ideal to directly extract mycobacterial DNA from clinical isolates for WGS to predict drug resistance. However, the high host DNA content and low microbial genome content are major obstacles to this analysis. In addition, the enriched microorganisms in respiratory specimens can also produce a large amount of extracellular DNA from biofilms or dead cells. In this study, we used early Bactec MGIT 960-positive liquid cultures as the research object and could obtain relatively pure *Mycobacterium* cultures. These purer cultures can be used to directly extract DNA for WGS, which can shorten TAT and help in obtaining comprehensive pathogenic microorganism information. A previous study showed that samples sequenced from MGIT cultures had higher reference genome coverage than those obtained directly from sputum (15). In this study, we used WGS from early positive liquid (MGIT) cultures to predict the drug resistance of 182 clinical isolates using the TB-Profiler informatics platform.

Studies have shown that more than 96% of rifampin resistance can be attributed to mutations in the 81-bp region of *rpoB* (RRDR) (21). In this study, only 2 isolates had no mutation in the RRDR region, which was a mutation at *rpoB* 464. This mutation was also reported in a study in Thailand and Turkey (22, 23). The mutations at *rpoB* 450 were found in 59.7% (46/71) of rifampin genotype-resistant isolates, and they confer rifampin and rifabutin resistance (5). For isoniazid, mutations in 98.7% (75/76) of the isoniazid genotype-resistant isolates were found on *katG* and *fabG1*, demonstrating the usefulness of these two specific regions for predicting isoniazid resistance (14). A *katG* Ser315Thr mutation was present in 61 cases, of which 52 had MIC values of ≥2 μg/mL. *KatG* Ser315Thr is the most frequently detected mutation in isoniazid-resistant TB, and it has been previously shown to confer high-level isoniazid resistance (5, 24–26). For ethambutol, most ethambutol resistance-related genes are located on *embB* and in the *embA* upstream region (27). The most common mutations were *embB* Met306Val and Met306Ile, accounting for 25.0% (13/52) and 19.2% (10/52), respectively. Six ethambutol-susceptible isolates were found with a mutation in Met306Ile, which was also reported in other studies (28, 29). Consistent with the results of other studies, there are three resistance genes related to streptomycin: namely, *rrs*, *rpsL*, and *gid*, accounting for 69.2% (72/104), 52.9% (55/104), and 3.8% (4/104), respectively, in all

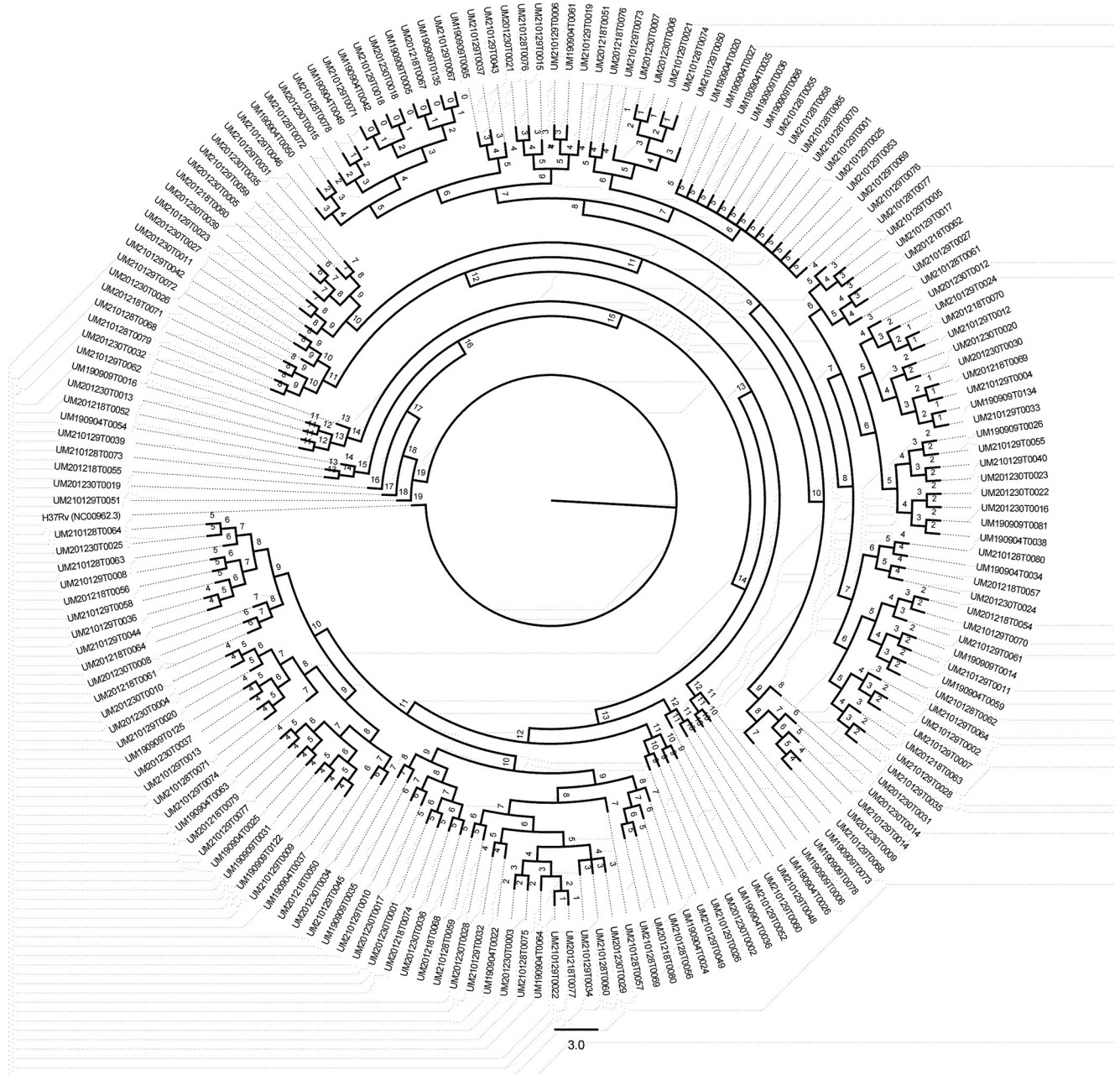

**FIG 1** Phylogenetic tree of 182 MTB isolates indicating drug resistance profiles and lineages with 81,622 high-confidence SNPs.

strains resistant to at least one drug. However, mutations in these three genes also occurred in streptomycin-susceptible isolates, which has also been reported in other studies (30, 31). In this study, 38 isolates with 888 G→A and 799 C→T mutations on *rrs* were phenotypically susceptible to streptomycin. Among them, the streptomycin MIC values were ≤0.25 μg/mL (*n* = 25), 0.5 μg/mL (*n* = 12) and 1 μg/mL (*n* = 1).The main mechanism of MTBC resistance to fluoroquinolones is the chromosomal mutations in the quinolone resistance determining region of *gyrA* or *gyrB (*32). In this study, the majority (89.7%, 35/39) of the isolates had mutations at 90 and 94 of *gyrA*, and very few at 89 and 91 of *gyrA* and 447, 461, 501 and 504 of *gyrB*. The mutations at positions 1401, 1402, and 1484 on *rrs* were considered to be associated with amikacin and kanamycin resistance, which appeared in 7, 59 and 47 isolates, respectively. In addition, 2 isolates had mutations in *eis* related to kanamycin resistance. For amikacin and kanamycin, 47

isolates had mutations of 1,402 C→A and 1,484 G→T, and 9 isolates only had mutations of 1,402 C→A, in *rrs*. Among them, all isolates were susceptible to amikacin, with a MIC value of ≤1 $\mu$g/mL, and 55 isolates were susceptible to kanamycin, with a MIC of ≤2.5 $\mu$g/mL. Therefore, 1,402 C→A and 1,484 G→T on *rrs* may not be associated with amikacin and kanamycin resistance, and this need to be further verified. Many studies have shown that there is cross-resistance between ethionamide and isoniazid, and mutations in *fabG1* are also the major mechanism of ethionamide resistance (33–37). For para-aminosalicylic acid, mutations of *thyA* and a flavin-based thymidylate synthase *thyX* appeared in 3 and 2 isolates, respectively, in this study, which has also been reported in other studies (38, 39). Known gene mutations related to cycloserine resistance were not found in this study (40–42).

In this study, we found that the sensitivity and specificity of WGS in predicting drug resistance against rifampin, isoniazid, ethambutol, streptomycin, moxifloxacin, ofloxacin, and amikacin were more than 80.00%, suggesting that WGS from early positive liquid (MGIT) cultures is a promising approach for predicting resistance to these drugs. For rifampin, isoniazid, moxifloxacin, ofloxacin and para-aminosalicylic acid, the predicted AST results using WGS were in good agreement with those of phenotypic AST; all greater than 90%, consistent with previously reported resistance profiles (13, 29, 43–45). Regarding ethambutol, this study has shown that the specificity of WGS in predicting the resistance of ethambutol was lower than that of other drugs. It was not uncommon for mutations such as Met306Ile and Gln497Arg on *embB* and −12C/T on *embA* to be found in resistant and susceptible isolates of ethambutol. This may be because phenotypic AST is less reliable in the case of ethambutol (46, 47). It is worth noting that, if mutations of 1,402 C→A and 1,484 G→T on *rrs* are not considered amikacin- and kanamycin-resistance genes, and 888 G→A and 799 C→T on *rrs* are not considered streptomycin-resistance genes, then the concordances of WGS-based AST and phenotypic AST could reach 100%, 99.45%, and 93.96%, respectively. The sensitivity of WGS in predicting ethionamide, para-aminosalicylic acid, and cycloserine resistance is low, possibly due to the unreliable results of phenotypic drug susceptibility testing and the remaining obstacles in the mechanisms of drug resistance (36, 48–52).

WGS from early positive liquid (MGIT) cultures has significant advantages over traditional phenotypic and molecular-based AST. First, WGS can quickly obtain complete drug-resistance mutations, which can lead to appropriate treatments. WHO released the first catalogue of MTBC mutations and their associations with drug resistance in June 2021, which seeks to support TB laboratories around the world in interpreting genome sequencing results (53). Second, for laboratories that do not have a biosafety level of II, it is easier to obtain drug susceptibility results through WGS. Third, in addition to predicting drug susceptibility, WGS can perform other studies, such as epidemiology. Finally, direct WGS using early positive liquid (MGIT) cultures instead of solid cultures can reduce TAT to a few weeks.

Our study has several limitations. First, the antituberculosis drugs involved in this study were limited, including pyrazinamide, capreomycin, and linezolid. In particular, the first-line drug pyrazinamide was not included in this study; traditional phenotypic pyrazinamide susceptibility testing is not routinely done because it must be performed under harsh acidic conditions (54). Second, the results for amikacin, kanamycin, ethionamide, para-aminosalicylic acid, and cycloserine may be biased, because the drug resistance rate was low, and the number of drug-resistant isolates included in the study was small. Third, the study was conducted in the clinical laboratory department, and not all of the whole-genome sequencing data were communicated to clinicians and patients. In future studies, we can use the advantage of WGS in shortening TAT to predict drug resistance after a comprehensive clinical evaluation.

In conclusion, WGS from early positive liquid (MGIT) cultures could be a promising approach to predict resistance to antituberculosis drugs, especially rifampicin, isoniazid, moxifloxacin, and ofloxacin. It is worth noting that the association of some

mutations with drug resistance still needs further study, such the mutations 1,402 C→A and 1,484 G→T on *rrs*.

## MATERIALS AND METHODS

**Study population.** A total of 211 clinical isolates were enrolled between 1 August 2019 and 30 October 2020 in the clinical laboratory department of Shanghai Pulmonary Hospital (SPH), affiliated with the Tongji University School of Medicine. SPH is a modern tertiary specialist hospital and the earliest TB prevention, detection, treatment, and research institution in China. All clinical isolates were isolated from culture-positive MTBC patients in Shanghai that were diagnosed and treated at SPH.

**MTB culture and phenotypic AST.** Laboratory examination was performed in the clinical laboratory of SPH, an ISO 15189-accredited laboratory specializing in MTBC detection and equipped with a full set of *Mycobacterium*-detection facilities. Clinical specimens, including sputum and bronchoalveolar lavage fluid, were collected from TB patients and treated with a 2% NaOH-*N*-acetyl-ʟ-cysteine (NaOH-NALC) solution. Liquid cultures were carried out according to relevant guidelines using a Bactec MGIT 960 instrument (Becton, Dickinson and Company; Cockeysville, MD, USA) in accordance with relevant guidelines. All MTB isolates were validated by both a growth test on p-nitrobenzoic acid containing medium (Baso, Zhuhai, China) and an MBP 64 antigen detection kit (Genesis, Hangzhou, China). Non-tuberculosis mycobacteria (NTM) were excluded. The 3-mL Bactec MGIT 960-positive liquid cultures were used to extract DNA for WGS, and the remaining positive cultures were switched to LJ medium (Baso, Zhuhai, China).

Colonies were taken from LJ medium and suspended in sterile saline containing 0.2% Tween and glass beads for AST. AST was performed using a MycoTB system (MYCOTB; Trek Diagnostic Systems, Thermo Fisher Scientific, USA). All steps were performed by trained and specialized staff in a biosafety cabinet in accordance with the relevant guidelines. The reference strain H37Rv was used for quality control once a month or for each new batch of susceptibility kit. The critical concentrations of MycoTB were 2.0 $\mu$g/mL for streptomycin, 0.2 $\mu$g/mL for isoniazid, 1.0 $\mu$g/mL for rifampin, 5.0 $\mu$g/mL for ethambutol, 4.0 $\mu$g/mL for amikacin, 2.0 $\mu$g/mL for ofloxacin, 25.0 $\mu$g/mL for cycloserine, 5.0 $\mu$g/mL for ethionamide, 5.0 $\mu$g/mL for kanamycin, 0.5 $\mu$g/mL for moxifloxacin, 2.0 $\mu$g/mL for aminosalicylic acid, and 0.5 $\mu$g/mL for rifabutin, respectively, according to CLSI M24-A2 and FDA-approved standards for AST.

**WGS sequencing and analysis.** The 3 mL Bactec MGIT 960 positive liquid cultures were used, and genomic DNA was extracted for sequencing using the cetyltrimethylammonium bromide (CTAB) method of DNA purification. Each extracted DNA was quantified by a Qubit 2.0 Fluorometer (Invitrogen, Carlsbad, CA, USA). Sequencing library preparations were constructed following the manufacturer's protocols (Illumina TruSeq DNA Nano Library Prep Kit). Next, libraries with different indices were multiplexed and loaded on an Illumina HiSeq instrument according to the manufacturer's instructions (Illumina, San Diego, CA, USA). Sequencing was carried out using a 2 × 150 paired-end (PE) configuration. The sequence reads were aligned to the reference strain MTB H37Rv (GenBank accession no. NC_000962.3). For sequencing depths of less than 20× or genome coverage of less than 90%, the sequencing quality was considered unqualified and not included in the study. All mutations were identified using the TB-Profiler informatics platform (https://github.com/jodyphelan/TBProfiler) under the default parameters. The results for each isolate were saved in text-format files, and only the SNP sites with frequencies greater than 10% were used to predict resistance.

**Phylogeny construction.** After the quality control process, all of the genome sequences enrolled were analyzed by Snippy software (https://github.com/tseemann/snippy, version 4.6.0) using the default parameters. The genome sequence of *Mycobacterium tuberculosis* H37Rv (NC_000962.3) was used as the reference to identify core SNPs, and then the SNP loci which were either found in the PE/PPE/PGRS gene region or were drug resistence-related were filtered out. The filtered core SNP sites were used for building the phylogeny tree using the FastTree software (http://www.microbesonline.org/fasttree/, version 1.4.4).

**Statistical analysis.** Statistical analysis was performed using MedCalc software. Sensitivity, specificity, PPV, and NPV of WGS were calculated with a 95% confidence interval using phenotypic AST as the gold standard.

**Data availability.** The WGS data for these isolates are available in the NCBI SRA database, under accession number PRJNA806507 (https://www.ncbi.nlm.nih.gov/bioproject/PRJNA806507). The accession numbers are listed in Data Set S2.

## SUPPLEMENTAL MATERIAL

Supplemental material is available online only.

**SUPPLEMENTAL FILE 1**, PDF file, 0.1 MB.

**SUPPLEMENTAL FILE 2**, XLSX file, 0.01 MB.

## ACKNOWLEDGMENTS

This project was supported through grants from the Shanghai Clinical Research Center for Infectious Disease (Tuberculosis) (grant no. 19MC1910800).

X.W. and F.Y. designed the study. X.W. wrote the manuscript. X.W. and F.Y. modified the manuscript. G.T. and X.W. performed the statistical analysis. J.Y. and Y.G. performed laboratory examination. W.S., H.S., and F.Y. supervised the project.

We declare no conflicts of interest.

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
