## [Reviewer comments · Microbiology Spectrum]

Microbiology Spectrum

Use of whole-genome sequencing to predict Mycobacterium tuberculosis complex drug resistance from early positive liquid cultures

Xiaocui Wu, Guangkun Tan, Wei Sha, Haican Liu, Jinghui Yang, Yinjuan Guo, Xin Shen, Zheyuan Wu, Hongbo Shen, and Fangyou Yu

Corresponding Author(s): Fangyou Yu, Shanghai Pulmonary Hospital, School of Medicine, Tongji University

Review Timeline:

Submission Date:	December 5, 2021
Editorial Decision:	January 11, 2022
Revision Received:	March 3, 2022
Accepted:	March 3, 2022

Editor: Jeanette TEO

Reviewer(s): The reviewers have opted to remain anonymous.

Transaction Report:

DOI: <https://doi.org/10.1128/spectrum.02516-21>

January 11, 2022

Dr. Fangyou Yu
Shanghai Pulmonary Hospital, School of Medicine, Tongji University
Department of Clinical Laboratory
Shanghai
China

Re: Spectrum02516-21 (Use of whole-genome sequencing to predict Mycobacterium tuberculosis complex drug resistance from early positive liquid cultures)

Dear Dr. Fangyou Yu:

Link Not Available

Sincerely,

Jeanette TEO

Journals Department
Reviewer comments:

Reviewer #1 (Public repository details (Required)):

The authors whole-genome sequenced 182 MTBC isolates including resistant isolates. In addition, they found some resistant isolates containing non-common resistant mutations. It will be helpful to share this data with the scientific community in order to improve the current diagnostic methods.

Reviewer #1 (Comments for the Author):

This interesting manuscript by Wu et al. uses WGS to predict antimicrobial resistance within MTBC clinical isolates. Although the methodology is accurate and the main purpose of the manuscript is clear. The manuscript could be further improved by addressing the points as follows.

Major comments:

The authors predicted genomically the resistance for many antimycobacterial drugs (including second-line drugs). However, it is

surprising that resistance to all first-line antibiotics, such as pyrazinamide, has not been tested. Do the authors explain why?. In addition, they predicted other non-common drugs such as para-aminosalicylic acid and cycloserine resistance, which are drugs used in XDR MTBC cases. These resistant cases (individuals) were included in the study?, do they know this before the study or during the study?. It will be interesting to add this clinical information in the manuscript, because one of the main advantages of using WGS as a prediction tool is the reduced time to detect these cases.

The authors used the public platform "TB profiler" for the genomically resistance prediction, which is a good approach. However, do the authors use other public platforms such as "Mykrobe predictor" or "PhyResSe"?. It is known that not all platforms have the same catalog of mutations. It may be interesting to make a comparison with different platforms, especially to resolve inconsistencies between phenotypic and genotypic resistance.

The authors have to describe in detail the methodology they followed to detect resistance variants (such as SNPs). Did they use a certain frequency?. Is very important to know the frequency of some mutations in each isolate, especially those not so common mutations. For instance, with this information, the evolution of resistant populations in an isolate can be seen. In clinical terms, this can help to redirect or change a treatment. They use the term high-confidence SNP, but it is not described in the methodology. In addition, do they use INDEL variants to predict resistance?. If so, how do they treat them?

A conclusion of the manuscript is that WGS reduces resistance detection time and that it is a better option than phenotypic diagnosis. Another point in favor of this technology would be to know the cost of this methodology with respect to the gold standard. Is it feasible? Particularly in this specific setting. These methods are implemented in the local surveillance system?. The authors could include this information in the discussion section.

Minor comments:

Introduction:

Lines 46, 48: Add the appropriate references.

Methods:

Line 110: WGS sequencing and analysis

Please add the methodology followed for the detection of high-confidence SNP variants.

Is the WGS-based data generated (raw fastqs) available in a public repository? If so, please add their accession numbers of each isolate in a table.

Line 123: Phylogeny reconstruction

Line 126: Please check that the scientific names are well written through the manuscript.

Line 128: How many SNPs they used for tree reconstruction?. Which reconstruction method they used for the tree inference?

Results:

Lines 141-144: Please check the numbers in the text and the tables. it has to see consistency in the numbers throughout the manuscript.

Line 145: Define MDR-TB within the manuscript.

Line 148: Please check the numbers in the text and the tables. it has to see consistency in the numbers throughout the manuscript.

Lines 193-214: It will be more clear to understand whether the authors summarize this information into a table.

Line 223: Which is a high-confidence SNP? Please describe this in the methods section.

Discussion:

Lines 245-246: Please be consistent with the mutation nomenclature through the whole manuscript.

Tables and figures:

Table 2: Please add the loci position for each mutation you detect as well as their respective amino acid change. It will be more clear to understand.

Figure 1: Please add the tree scale used as well as the bootstrap value for each branch.

Reviewer #2 (Public repository details (Required)):

Please submit WGS sequencing data to GeneBank.

Reviewer #2 (Comments for the Author):

The paper evaluates the performance of whole-genome sequencing (WGS) from early positive liquid cultures for predicting Mycobacterium tuberculosis complex (MTBC) drug resistance. The innovation of this paper lies in the selection of early positive

liquid culture as the detection object, which shortens the sample turn-around time (TAT) compared with solid culture. However, a few minor revisions are listed below.

1. Due to the great diversity of genetic background and complex mechanism of drug resistance and evolution of clinical strains, it is difficult to find reliable drug resistance mutations by using the correlation analysis of phenotype and genotype. For isolates with inconsistent phenotypic and genotypic drug resistance, further validation, such as in vitro induction test combined with whole genome sequencing, should be performed.
2. In this study, many high-confidence SNPs were found, which can be used to analyze the expression of downstream genes that these SNPs may regulate. Are these genes related to cell wall biosynthesis, drug pump, transcriptional regulation, etc.?
3. The final conclusion is too general and optimistic. It should be summarized which mutation sites in this study can be used to predict drug resistance and which mutation sites are not yet reliable.

Staff Comments:

Preparing Revision Guidelines

Please return the manuscript within 60 days; if you cannot complete the modification within this time period, please contact me. If you do not wish to modify the manuscript and prefer to submit it to another journal, please notify me of your decision immediately so that the manuscript may be formally withdrawn from consideration by Microbiology Spectrum.

This interesting manuscript by Wu et al. uses WGS to predict antimicrobial resistance within MTBC clinical isolates. Although the methodology is accurate and the main purpose of the manuscript is clear. The manuscript could be further improved by addressing the points as follows.

Major comments:

The authors predicted genomically the resistance for many antimycobacterial drugs (including second-line drugs). However, it is surprising that resistance to all first-line antibiotics, such as pyrazinamide, has not been tested. Do the authors explain why?. In addition, they predicted other non-common drugs such as para-aminosalicylic acid and cycloserine resistance, which are drugs used in XDR MTBC cases. These resistant cases (individuals) were included in the study?, do they know this before the study or during the study?. It will be interesting to add this clinical information in the manuscript, because one of the main advantages of using WGS as a prediction tool is the reduced time to detect these cases.

The authors used the public platform "TB profiler" for the genomically resistance prediction, which is a good approach. However, do the authors use other public platforms such as "Mykrobe predictor" or "PhyResSe"?. It is known that not all platforms have the same catalog of mutations. It may be interesting to make a comparison with different platforms, especially to resolve inconsistencies between phenotypic and genotypic resistance.

The authors have to describe in detail the methodology they followed to detect resistance variants (such as SNPs). Did they use a certain frequency?. Is very important to know the frequency of some mutations in each isolate, especially those not so common mutations. For instance, with this information, the evolution of resistant populations in an isolate can be seen. In clinical terms, this can help to redirect or change a treatment. They use the term high-confidence SNP, but it is not described in the methodology. In addition, do they use INDEL variants to predict resistance?.If so, how do they treat them?

A conclusion of the manuscript is that WGS reduces resistance detection time and that it is a better option than phenotypic diagnosis. Another point in favor of this technology would be to know the cost of this methodology with respect to the gold standard. Is it feasible? Particularly in this specific setting. These methods are implemented in the local surveillance system?. The authors could include this information in the discussion section.

Minor comments:

Introduction:

Lines 46, 48: Add the appropriate references.

Methods:

Line 110: WGS sequencing and analysis

Please add the methodology followed for the detection of high-confidence SNP variants.

Is the WGS-based data generated (raw fastqs) available in a public repository? If so, please add their accession numbers of each isolate in a table.

Line 123: Phylogeny reconstruction

Line 126: Please check that the scientific names are well written through the manuscript.

Line 128: How many SNPs they used for tree reconstruction?. Which reconstruction method they used for the tree inference?

Results:

Lines 141-144: Please check the numbers in the text and the tables. it has to see consistency in the numbers throughout the manuscript.

Line 145: Define MDR-TB within the manuscript.

Line 148: Please check the numbers in the text and the tables. it has to see consistency in the numbers throughout the manuscript.

Lines 193-214: It will be more clear to understand whether the authors summarize this information into a table.

Line 223: Which is a high-confidence SNP? Please describe this in the methods section.

Discussion:

Lines 245-246: Please be consistent with the mutation nomenclature through the whole manuscript.

Tables and figures:

Table 2: Please add the loci position for each mutation you detect as well as their respective amino acid change. It will be more clear to understand.

Figure 1: Please add the tree scale used as well as the bootstrap value for each branch.

Reviewer comments:

Reviewer #1 (Public repository details (Required)):

The authors whole-genome sequenced 182 MTBC isolates including resistant isolates. In addition, they found some resistant isolates containing non-common resistant mutations. It will be helpful to share this data with the scientific community in order to improve the current diagnostic methods.

Response: Thank you very much for your comments. The WGS data of these isolates have been submitted to the NCBI SRA database, under accession number PRJNA806507 (<https://www.ncbi.nlm.nih.gov/sra/PRJNA806507>).

Reviewer #1 (Comments for the Author):

This interesting manuscript by Wu et al. uses WGS to predict antimicrobial resistance within clinical isolates. Although the methodology is accurate and the main purpose of the manuscript is clear. The manuscript could be further improved by addressing the points as follows.

Major comments:

The authors predicted genomically the resistance for many antimycobacterial drugs (including second-line drugs). However, it is surprising that resistance to all first-line antibiotics, such as pyrazinamide, has not been tested. Do the authors explain why?. In addition, they predicted other non-common drugs such as para-aminosalicylic acid and cycloserine resistance, which are drugs used in XDR MTBC cases. These resistant cases (individuals) were included in the study?, do know this before the study or during the study?. It will be interesting to add this clinical in the manuscript, because one of the main advantages of using WGS as a prediction tool is the reduced time to detect these cases.

Response: As what we have added in the discussion section, traditional phenotypic pyrazinamide susceptibility is not routinely performed due to it needs to be performed under harsh acidic conditions. And because this is only a pre-research project which was conducted in the department of clinical laboratory, and the consistency of WGS data and the phenotypic resistance was still needed to be proved, so the predicted data was not communicated to the clinic branch to identify the XDR cases. But we hope in the future, after the comprehensive clinical evaluation, will use the whole genome sequencing data to predict drug resistance and to shorten the diagnostic duration.

The authors used the public platform "TB profiler" for the genomically resistance prediction, is a good approach. However, do the authors use other public platforms such as "Mykrobe predictor" or "PhyResSe"?. It is known that not all platforms have the same catalog of mutations. may be interesting to make a comparison with different platforms, especially to resolve inconsistencies between phenotypic and genotypic resistance.

Response: Thank you very much for your comments. As showed in the previous studies that TB profiler has good agreement when comparing the WGS-based

tools against the culture-based AST, so we adopted TB profiler as the WGS-based tool in this study to identify the mutations preserved in the sample genomes. And your suggestion has really helped us a lot, we will do the evaluation between different platforms and softwares by comparing the detailed mutation profiles in the future.

The authors have to describe in detail the methodology they followed to detect resistance variants (such as SNPs). Did they use a certain frequency?. Is very important to know the frequency of some mutations in each isolate, especially those not so common mutations. For instance, with this information, the evolution of resistant populations in an isolate can be seen. In clinical terms, this can help to redirect or change a treatment. They use the term high-confidence SNP, but it is not described in the methodology. In addition, do they use INDEL variants to predict resistance?. If so, how do they treat them?

Response: Thank you very much for your comments. As what we have described in the methods section, all the mutations were identified by the TB-Profiler informatics platform (<https://github.com/jodyphelan/TBProfiler>) under the default parameters. The results of each isolate were saved in text format files, and only the SNP sites that have a frequency above 10% were used to predict resistance. And we have added a supplementary table titled "The amino acid mutations and frequency associated with drug resistance".

A conclusion of the manuscript is that WGS reduces resistance detection time and that it is a better option than phenotypic diagnosis. Another point in favor of this technology would be to know the cost of this methodology with respect to the gold standard. Is it feasible? Particularly in this specific setting. These methods are implemented in the local surveillance system?. The authors could include this information in the discussion section.

Response: Thank you very much for your comments. The cost of WGS for one sample (PE250, 100x depth, 95% coverage) is about \$60, which is slightly higher than that of culture-based AST for 8 or 12 commonly used anti-TB drugs. But could supply much more information for estimating the drug sensibility according to the reported mutation markers.

Currently, the use of WGS for predicting TB resistance in China is not routinely used clinically due to its difficulty in standardization, interpretation of results, etc. With the sharing of genome-wide data, the in-depth research of drug resistance mechanisms and the development of sequencing technology, WGS may completely change AST in conventional microbiology laboratories. We have added the above in the discussion section.

Minor comments:

Introduction:

Lines 46, 48: Add the appropriate references.

Response: We have added it.

Methods:

Line 110: WGS sequencing and analysis

Please add the methodology followed for the detection of high-confidence SNP variants.

Response: We have described this in the “Phylogeny construction” section. All the core SNPs were identified by using the Snippy software (<https://github.com/tseemann/snippy>, version: 4.6.0) use the default parameters according to the reference genome (H37Rv, NC_00962.3). The SNP loci which located in the PE/PPE/PGRS gene region or drug resistance related were filtered out. And then the lefted SNPs were used for further analysis.

Is the WGS-based data generated (raw fastqs) available in a public repository? If so, please add their accession numbers of each isolate in a table.

Response: Thank you very much for your comments. The WGS data of these isolates have been submitted to the NCBI SRA database, under accession number PRJNA806507 (<https://www.ncbi.nlm.nih.gov/sra/PRJNA806507>). The accession numbers of each isolate and corresponding URLs are showed in Supplementary table 2.

Line 123: Phylogeny reconstruction

Response: We have described this in the “Phylogeny construction” section. All the core SNPs were identified by using the Snippy software (<https://github.com/tseemann/snippy>, version: 4.6.0) use the default parameters according to the reference genome (H37Rv, NC_00962.3). The SNP loci which located in the PE/PPE/PGRS gene region or drug resistance related were filtered out. And then the lefted SNPs were used for further analysis.

Line 126: Please check that the scientific names are well written through the manuscript.

Response: We have checked the scientific names.

Line 128: How many SNPs they used for tree reconstruction?. Which reconstruction method they used for the tree inference?

Response: Thank you very much for your comments. 81,622 high-confidence SNPs that we refer to in the "Phylogenetic distribution of drug resistance" of the results. The core SNPs were identified by Snippy, and filtered by local Perl script to remove the SNP loci which were located in the PE/PPE/PGRS regions or drug resistance related. And then the aligned SNPs from each genome and ML method were used to build the Phylogeny tree.

Results:

Lines 141-144: Please check the numbers in the text and the tables. it has to see consistency in the numbers throughout the manuscript.

Response: We have checked it.

Line 145: Define MDR-TB within the manuscript.

Response: We have added it.

Line 148: Please check the numbers in the text and the tables. it has to see consistency in the numbers throughout the manuscript.

Response: We have checked the numbers in the text and the tables.

Lines 193-214: It will be more clear to understand whether the authors summarize this information into a table.

Response: Thank you very much for your comments. we have added table 4 titled "The amino acid mutations and frequency associated with drug resistance".

Line 223: Which is a high-confidence SNP? Please describe this in the methods section.

Response: We have described this in the "Phylogeny construction" section. All the core SNPs were indentified by using the Snippy software (<https://github.com/tseemann/snippy>, version: 4.6.0) use the default parameters acording to the reference genome (H37Rv, NC_00962.3). The SNP luci which located in the PE/PPE/PGRS gene region or drug resistance related were filtered out. And then the lefted SNPs were used for further analysis.

Discussion:

Lines 245-246: Please be consistent with the mutation nomenclature through the whole manuscript.

Response: We have checked it.

Tables and figures:

Table 2: Please add the loci position for each mutation you detect as well as their respective amino acid change. It will be more clear to understand.

Response: Thank you very much for your comments. We have added a supplementary table titled "The amino acid mutations and frequency associated with drug resistance ".

Figure 1: Please add the tree scale used as well as the bootstrap value for each branch.

Response: We have updated a new figure.

Reviewer #2 (Public repository details (Required)):Please submit WGS sequencing data to GeneBank.

Response: Thank you very much for your comments. We have submitted the data to SRA database.

Reviewer #2 (Comments for the Author):

The paper evaluates the performance of whole-genome sequencing (WGS) from early positive liquid cultures for predicting Mycobacterium tuberculosis complex (MTBC) drug resistance. The innovation of this paper lies in the selection of early positive liquid culture as the detection object, which shortens the sample turn-around time (TAT) compared with solid culture. However, a few minor revisions are listed below.

1. Due to the great diversity of genetic background and complex mechanism of drug resistance and evolution of clinical strains, it is difficult to find reliable drug resistance mutations by using the correlation analysis of phenotype and genotype. For isolates with inconsistent phenotypic and genotypic drug resistance, further validation, such as in vitro induction test combined with whole genome sequencing, should be performed.

Response: Thank you very much for your comments. Since this study was designed to evaluate the performance of WGS from early positive liquid cultures for predicting MTBC drug resistance based on the reported SNPs, further studies are needed, such as for the isolates that with inconsistent phenotypic and genotypic drug resistance, which will be further discussed in future research reports.

2. In this study, many high-confidence SNPs were found, which can be used to analyze the expression of downstream genes that these SNPs may regulate. Are these genes related to cell wall biosynthesis, drug pump, transcriptional regulation, etc.?

Response: The aim of this project was to evaluate the performance of WGS from early positive liquid cultures for predicting MTBC drug resistance based on the reported SNP sites, we did not do further analysis, such as the effect of gene's function by the mutations which were described in this manuscript.

3. The final conclusion is too general and optimistic. It should be summarized which mutation sites in this study can be used to predict drug resistance and which mutation sites are not yet reliable.

Response: Thanks for your suggestion, we have modified the description. In conclusion, WGS from early positive liquid (MGIT) cultures could be a promising approach to predict resistance to anti tuberculosis drugs, especially for rifampicin, isoniazid, moxifloxacin, and ofloxacin. It is worth noting that the association of some mutations and drug resistance still needs further study, such as mutations of 1402c>a and 1484g>t on *rrs*.

March 3, 2022

Dr. Fangyou Yu
Shanghai Pulmonary Hospital, School of Medicine, Tongji University
Department of Clinical Laboratory
Shanghai
China

Re: Spectrum02516-21R1 (Use of whole-genome sequencing to predict Mycobacterium tuberculosis complex drug resistance from early positive liquid cultures)

Dear Dr. Fangyou Yu:

Your manuscript has been accepted, and I am forwarding it to the ASM Journals Department for publication. You will be notified when your proofs are ready to be viewed.

Sincerely,

Jeanette TEO
Editor, Microbiology Spectrum

Journals Department
Supplemental Material: Accept
Supplemental Dataset: Accept